# Investigation of the Detailed AMPylated Reaction Mechanism for the Huntingtin Yeast-Interacting Protein E Enzyme HYPE

**DOI:** 10.3390/ijms22136999

**Published:** 2021-06-29

**Authors:** Meili Liu, Zhe Huai, Hongwei Tan, Guangju Chen

**Affiliations:** 1Department of Chemistry, Beijing Normal University, Beijing 100875, China; meililiu@berkeley.edu (M.L.); gjchen@bnu.edu.cn (G.C.); 2Department of Chemical and Biomolecular Engineering, University of California, Berkeley, CA 94720, USA; 3State Key Laboratory of Precision Spectroscopy, School of Physics and Electronic Science, East China Normal University, Shanghai 200062, China; huaizhe123@163.com

**Keywords:** density functional theory, AMPylation, Fic-mediated, HYPE

## Abstract

AMPylation is a prevalent posttranslational modification that involves the addition of adenosine monophosphate (AMP) to proteins. Exactly how Huntingtin-associated yeast-interacting protein E (HYPE), as the first human protein, is involved in the transformation of the AMP moiety to its substrate target protein (the endoplasmic reticulum chaperone binding to immunoglobulin protein (BiP)) is still an open question. Additionally, a conserved glutamine plays a vital key role in the AMPylation reaction in most filamentation processes induced by the cAMP (Fic) protein. In the present work, the detailed catalytic AMPylation mechanisms in HYPE were determined based on the density functional theory (DFT) method. Molecular dynamics (MD) simulations were further used to investigate the exact role of the inhibitory glutamate. The metal center, Mg^2+^, in HYPE has been examined in various coordination configurations, including 4-coordrinated, 5-coordinated and 6-coordinated. DFT calculations revealed that the transformation of the AMP moiety of HYPE with BiP followed a sequential pathway. The model with a 4-coordinated metal center had a barrier of 14.7 kcal/mol, which was consistent with the experimental value and lower than the 38.7 kcal/mol barrier of the model with a 6-coordinated metal center and the 31.1 kcal/mol barrier of the model with a 5-coordinated metal center. Furthermore, DFT results indicated that Thr518 residue oxygen directly attacks the phosphorus, while the His363 residue acts as H-bond acceptor. At the same time, an MD study indicated that Glu234 played an inhibitory role in the α-inhibition helix by regulating the hydrogen bond interaction between Arg374 and the P_γ_ of the ATP molecule. The revealed sequential pathway and the inhibitory role of Glu234 in HYPE were inspirational for understanding the catalytic and inhibitory mechanisms of Fic-mediated AMP transfer, paving the way for further studies on the physiological role of Fic enzymes.

## 1. Introduction

Posttranslational modifications (PTMs) of proteins are an important cellular mechanism and typically used to alter diverse functions and locations such as signaling, localization or protein–protein interactions. The introduction of protein PTMs is a tightly controlled and almost ubiquitous process [1,2,3,4,5]. Modifications by proteases, kinases, methylases, and acetylases have been explored extensively, and their misregulation is often associated with severe pathology, including autoimmune diseases or cancer [6,7]. 

One such PTM common to Fic proteins that has recently been gaining attention is AMPylation (also referred to adenylation). AMPylation was first discovered in the 1960s as a regulatory mechanism for controlling glutamine synthetase activity in *Escherichia coli* [8]. Afterward, it was found that bacterial effectors from *Vibrio parahaemolyticus* and *Histophilus somni* AMPylate Rho guanosine triphosphatases (GTPases) were found to exist in human host cells [9,10]. These bacterial effectors contain highly conserved Fic domains, and the HXFX(D/E)GNGRXXR sequence motif, which uses ATP to covalently add an adenosine monophosphate (AMP) moiety to target proteins, and the most common and stable form of AMPylation occurs on the hydroxyl group of threonine, serine, or tyrosine through a phosphodiester bond [11]. The presence of a Fic domain is essential for AMP transfer [12,13]. Fic proteins include two critical elements that are the catalytic loop for the enzyme activity, and an inhibitory α-helix (α_inh_) for some Fic domains [14,15,16]. Despite their abundance in bacteria, only one human protein AMPylator containing the signature Fic domain, named Huntingtin yeast partner E (HYPE), was discovered and exhibited AMPylation activity against Cdc42, Rac1, and BiP in vitro [10,17,18,19,20]. 

HYPE is present in most cell types and tissues, although at low levels. Furthermore, HYPE is assumed to interact with Huntingtin, a protein that when mutated, causes Huntington disease [21], a neurodegenerative disorder. The catalytic loop in HYPE proteins shares the general signature motif of Fic domains, HxFx(D/E)(A/G)N(G/K)R, represented in HYPE by the sequence HPF(I/V)DGNGRT(S/A)R. The critical His residue within the catalytic motif corresponds to His363 in the human HYPE [18,22]. Interestingly, the catalytic activity of a group of Fic enzymes is regulated through the presence of an inhibitory α-helix (α_inh_). The inhibitory helix (α_inh_) contains a common inhibitory signature, (S/T)xxxE(G/N), conserved in HYPE proteins as (T/S)V(A/G)IEN. Inhibitory glutamate, Glu234, from α-inh is positioned in the vicinity of the catalytic loop [20]. Conserved glutamate protrudes into the phosphate binding pocket of the catalytic site of Fic enzymes and thereby sterically and electrostatically obstructs ATP binding and phosphate positioning [20]. However, the mechanism of the release of inhibition is currently unknown. 

Most of the known enzymes that catalyze AMPylation are bacterial effectors that are secreted into infected cells to AMPylate small GTPases (Rho and Rab families), and cause damage to the host cells [9,23,24]. These bacterial effectors are considered to be potential new targets for drug discovery because AMPylation plays an extraordinary role in infection [25]. Well-studied examples of Fic domain proteins are, for instance, VopS [26], AnkX [27] and IbpA [28], which catalyze AMPylation of Rho GTPases disrupting the actin cytoskeleton of the host cell. Experimental studies have found that HYPE is highly similar to the VopS protein which lacks α_inh_ [18,26,29]. At the same time, BiP, one of the target proteins of HYPE, is a novel substrate for *Drosophila* mediated AMPylation. Furthermore, the experiments revealed that AMP attached to a single amino acid of BiP, number 518, a threonine residue [4].

Elucidating the mechanistic details of the AMPylation reaction has therefore been the subject of intensive research. Numerous experimental studies have been performed to explore the AMPylation reaction mechanisms of Fic proteins, such as the ping-pong mechanism [27,30,31] and sequential mechanism [26] (shown in Scheme 1). However, how HYPE achieves AMPylation remains an open question. Meanwhile, as a bifunctional enzyme of AMPylation and deAMPylation [1], detailed mechanistic studies of the role of HYPE in AMPylation may also help us understand the more complex problem of Fic protein’s deAMPylation functions and provide theoretical insights to find more new Fic proteins and perform more possible research on Fic proteins.

**Scheme 1 ijms-22-06999-sch001:**
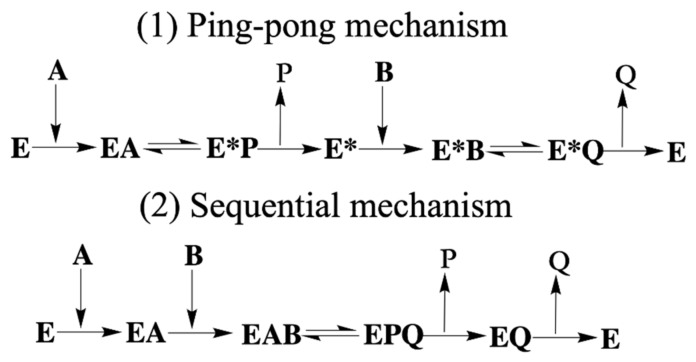
Two reaction mechanisms in the Fic protein. (**1**) Ping-pong mechanism [27,30,31]. (**2**) Sequential mechanism [26]. E represents enzyme. A and B represent the different reactant substrate. P and Q are the products. EA are the binary complex. EAB and EPQ mean ternary complex.

In the present study, by employing the hybrid density functional theory (DFT) method B3LYP [32], we explored the catalytic mechanism of BiP catalyzed by the HYPE enzyme in detail. All transition states and stationary points along the reaction path were located. Our calculation results indicate that the HYPE enzyme used for the BiP target protein follows a sequential pathway, and its metal center, Mg^2+^, is 4-coordinated. In addition, we further investigated the exact role of the inhibitory glutamate Glu234 in α_inh_ by using molecular dynamics (MD) simulation to clarify the experimental observation that the glutamate residue on the inhibitory helix plays a key inhibitory effect.

## 2. Results

### 2.1. AMPylation Mechanism of HYPE Complexed with BiP

Three possible geometries for the binding of the two proteins, the HYPE protein and substrate BiP protein, were investigated. The schematic structures and the computed free energy profiles of the active site geometries of the reactant for the AMPylation reaction are depicted in Figure 1. In Figure 1 Model-1, the coordination number of Mg(II) ions is six, and the ligands are P_α_-O, P_β_-O, P_γ_-O (ATP), HOH, a negatively charged residue (Asp367 from HYPE), and a positively charged residue (Lys516 from the substrate BiP protein). The 5-coordinated geometry is shown in Figure 1 Model-2, with the ligands P_α_-O, P_β_-O(ATP), HOH, a negatively charged residue (Asp367 from HYPE), and a positively charged residue (Lys516 from the substrate target protein, BiP). A Mg^2+^ ion in the E234A variant of HYPE bridges the α- and β-phosphates and is coordinated by the conserved Asp367 side chain. The conserved arginine at the C-terminal side of the Fic catalytic loop, Arg374, forms hydrogen bonds with the ribose ring and is also critical for binding γ-phosphate. Figure 1 Model-3 shows the geometries of 4-coordinated Mg(II) ions complexed with P_α_-O, P_β_-O, P_γ_-O (ATP) and Asp367.

The calculated energetic data are listed in Figure 1a–c. The 4-coordinated model is favorable, with an energy barrier of 14.7 kcal/mol, while the 5-coordinated model and hexacoordinated model experience TSs with energy barriers of 31.1 kcal/mol and 38.7 kcal/mol, respectively. Meanwhile, they are endothermic reactions for Model-1 and Model-2 (Figure 1a,b). An exothermic reaction easily occurs for Model-1 (Figure 1c). HYPE was the first human Fic protein, sequence analysis revealed that HYPE shared some similarities with VopS [33]. To easily understand the differences between the sequences of the HYPE protein and VopS protein, we made Appendix A. Key amino acids of the active site motif (His) and inhibitory motif (Glu) are highlighted in red and green, respectively. The Fic domain (represented by a gray circle) is essential for AMP transfer. Compared with VopS, HYPE has a Fic motif and α_inh_ in the Fic domain, while VopS has a Fic motif. The Glu234Ala mutation in our calculations uses the Ala234 residue to replace the key residue Glu234 in α_inh_. Therefore, we further compared our results with the kinetic constant measurements of VopS [26]. The apparent kinetic constants measured for the ATP of wild-type VopS [26] were 26 ± 1.0 s^−1^ (approximately 15 kcal/mol), which is consistent with the 14.7 kcal/mol constant obtained from Model-3.

Scheme 1 shows the details of the ping-pong mechanism and a sequential reaction mechanism. The largest feature is the ternary complex in the sequential mechanism. From this point on, three geometries in our calculations all supported the sequential mechanism (ternary complex (EAB), shown in Scheme 1(2)), indicating that the HYPE protein would interact with both ATP and the substrate BiP protein to allow transfer of the AMP group to the threonine residue. 

As listed in Table 1, three coordinated species, 6-coordinated, 5-coordinated and 4-coordinated species, were located and the coordinate details are shown in table S–S6. For inspection of the bond lengths in Table 1, r16, r17, r18, r19, r20, and r21 (Figure 1 Model-1, Model-2, and Model-3) represent the changes in the bond length of the coordination bond for the stationary points. The changes are small. r1(O_Thr518_-H_Thr518_), r2(N_His363_-H_Thr518_), r3(P_α_-O_Thr518_), and r7(P_α_-O_αβ_) are closely related to the AMPylation reaction process (see Figure 1 Model-1, Model-2, Model-3). We focus on the distances of r1, r2, r3 and r7 shown in Figure 2 in the reactant, TS and product. Notably, r1 and r2 are 1.74 Å and 1.90 Å in TS1 and TS2 and 2.10 Å and 2.10 Å in P1 and P2, respectively. In contrast, r1 and r2 are 1.02 Å in TS3 and 1.83 Å in P3. When going from the reactant to the product, experimental researchers proposed [34] that the His residue, as a general base, abstracts the proton from the hydroxyl group of the Thr residue by forming a N_His_-H_Thr_ bond, which is accompanied by the breakage of the O_Thr_-H_Thr_ bond. At the same time, the P_α_-O_αβ_ bond breaks down from the reactant to product states. Additionally, a P_α_-O_Thr518_ bond formed. It was involved in the 6-coordinated and 5-coordinated species in this work. Interestingly, His363 on Model-3 with magnesium coordinated by Asp367 and HOH606 interacts with β-phosphate, and the γ-phosphate of ATP acts as an acceptor to accept a proton from the hydroxyl group of the Thr518 residue of the BiP protein. Thus, our calculations predict that (a) the proton transfer processes in the two models of 6-coordination and 5-coordination are nearly complete in the TS and that (b) the formation of P_α_-O_Thr518_ and the breakage of O_Thr518_-H_Thr518_ occur after proton transfer. 

Additionally, in the 4-coordination model, proton transfer between His363 and Th518 is completed later, and the transition state is reached. In considering computed free energy profiles, later proton transfer (14.7 kcal/mol) is favored in the AMPylation reaction of HYPE. Furthermore, the electrostatic potentials (ESPs) [35] on the molecular vdW surface of the reactants for the three models are visualized in Appendix A. The yellow and magenta spheres represent the surface local maxima and minima of the ESP. The ESP values encircling P_α_ are positive for the 4-coordinated model (Appendix A), while the ESP values are negative for the 6-coordinated (Appendix A) and 5-coordinated models (Appendix A). This helps us understand the phenomenon that His363 acts as an acceptor instead of a general base in model-3 with 4-coordinated Mg^2+^ during the AMPylation process.

### 2.2. The Exact Role of Glu234 in AMPlation

Previous experimental evidence indicated that inhibitory glutamate in the inhibitory α-helix plays a vital role. To gain further insight into the detailed role of the inhibitory glutamate, we carried out a detailed analysis of its functional and structural role. As shown in Appendix A, during the 70 ns MD simulation, the average root-mean-square deviation (RMSD) of the whole protein complexes and the ATP molecule at approximately 5 Å in complexes with Glu234 and Glu234Ala converged well, indicating that stable models in HYPE complexed with BiP. Figure 3 depicts the interaction map of equilibrated structures for two geometries with Glu234 (Figure 3a, Glu234) and with Glu234Ala (Figure 3b, Ala234). We can identify important residues contributing to favorable interactions. The interaction map of Figure 3a tells us that Gly368, Asn369, Gly370 and Leu403 form stable hydrogen bonds with ATP molecules. The corresponding mutated structures in Figure 3b show that Arg374, Gly370, Arg371, Lys534, Gly368, Asn369, Tyr399 and Hie356 form stable hydrogen bonds with ATP molecules. Note that in Figure 3a, Arg371 formed two interactions with O_β_ and O_γ_, and Arg374 had only one interaction with O_β_. The interactions include salt bridge interactions and no Pi–Pi stacked interactions. In Figure 3b, Arg371 also interacts with O_β_ and O_γ_; at the same time, Arg374 has two interactions with O_γ_ (Figure 3b and Appendix A). In contrast, the interactions include Pi–Pi stacked interactions and no salt bridge interactions. The two structures are very similar so that the interaction maps of the two are also similar. However, the mutated structure (Figure 3b) is more stable according to the interaction map. Appendix A shows the number of hydrogen bonds between the ATP molecule and the surrounding residues. The average number of hydrogen bonds surrounded by ATP molecule is 9.3 for the mutated structure, Ala234, and 5.2 for the structure with Glu234. This further revealed that the ATP molecule has a more stable environment in the mutated structure, Ala234, than in the structure with Glu234.

The root-mean-squared fluctuation (RMSF) of C-α atoms was calculated for two geometries, the results of which are shown in Appendix A. The RMSF plots of the two complexes, except for several parts of the complexes, are similar, which indicates that the interactions between ATP and the surrounding residues of the complexes are similar. For example, we can find a large conformational fluctuation around residue 374. To clarify the difference in details, we checked the geometries using the equilibrium structures. Figure 4a shows the superimposition of the active sites in Glu234 (Figure 4b) and Glu234Ala (Figure 4c). From the geometries, we know that Arg374 forms a hydrogen bond with Glu234 when Glu234 exists (Figure 4b). When Glu234 is replaced by Ala234, the O_γ_-P of ATP molecules has a hydrogen interaction with Arg374 (Figure 4c and Appendix A). Therefore, we can conclude that the hydrogen interaction between the Arg374 and O_γ_-P of ATP molecules favors the transformation of the AMP group. Glu234 inhibited the AMPylation of the HYPE protein with the BiP protein by controlling the hydrogen interaction of the Arg374 residue. Meanwhile, the position of the inhibitory glutamate (Glu234) in the structure of wild-type HYPE is consistent with its role in competing with the Arg374/γ-phosphate interaction (Figure 4). Arginine enters the active site of HYPE to stabilize the transition state of the phosphoryl transfer reaction.

## 3. Discussion

DFT calculations were employed to systematically study the mechanism by which HYPE catalyzed the targeted BiP protein. The calculation results suggest that HYPE utilized a sequential (ternary complex) mechanism to AMPylate the target BiP protein. The His363 residue played a critical role in the AMPylation reaction to accept the proton from Thr518 instead of abstracting the proton of Thr518 in the substrate BiP protein. The structural comparison of TS1, TS2 and TS3 (Figure 2) in this work also confirmed the importance of proton transmission in His363 residues. In addition, we discovered that Mg^2+^ coordinated with ATP molecules in the HYPE protein via 4-coordination, and the corresponding energy barrier is 14.7 kcal/mol, which is consistent with experimental value and favors the AMPylation of the HYPE protein complexed with the targeted BiP protein.

Experimental researchers have reported that Glu234 plays a key role [20] in inhibiting the AMPylation reaction of the HYPE protein with the BiP protein. In this work, a series of classical MD simulations of HYPE complexed with BiP provided insight into the exact role of inhibitory glutamate in forming hydrogen bond interactions with Arg374, resulting in an unfavorable geometry for the AMPylation reaction when glutamate exists. When glutamate was mutated into alanine, the Arg374 salt bridge ion paired with Arg374 would be broken so that O_γ_-P would have a stable hydrogen interaction (Figure 4c and Appendix A) with Arg374, which favors the AMPylation reaction.

A different hydrogen interaction environment is mainly the possible reason mentioned several times in the experiment [20,33]. Our results also have a better understanding for the regulation process of HYPE protein and BiP protein. As shown in Figure 5a,b, Glu234 affects the coordination number of the metal center by changing the hydrogen interaction mode between amino acid 234 and O_γ_-P or Arg374 residue, thereby changing the charge and interaction environment of the active center, and ultimately controlling the transfer of protons in His363, thereby controlling the AMPylation reaction. 

This work provided the theoretical evidence for the exact role of the inhibitory glutamate. And HYPE as the first human Fic protein, HYPE-catalyzed mechanism provides a new understanding of the Fic proteins. Our findings also provide a basis to consider further possible alternative cofactors of HYPE and distinct modes of target-recognition. 

## 4. Computational Details

### 4.1. Models

Cluster models including chain A of the HYPE (PDB Code: 4U07) [20] crystal structure and the targeted BiP protein (PDB Code: 5O4P) [36] were built based on two X-ray crystal structures (Figure 6 and Appendix A). First, the 30 putative orientations of BiP docked to HYPE were obtained by Z-dock software [37]. Geometry analysis and experimental information [1,38] supported that Thr518 [2,4,5,20,33,38,39] as the structurally preferred modification site. Taking the positions of active site His363 [4,5,20,33] and Thr518 as the criteria, the rational docking structures were selected as the starting geometry. Molecular dynamics (MD) simulations for the two geometries, Glu234 and Glu234Ala variants, of the HYPE complex with BiP were performed. The first geometry (Figure 4b) used wild-type HYPE with the Glu234 of the inhibitory α-helix. His363 was designated HiD and the potential proton acceptor in the AMPylation reaction. The second geometry (Figure 4c) was built based on the first geometry, with Glu234 mutated to an alanine. All water molecules were removed. The protonation states of the protein were determined using PropKa [40] and visual inspection. After adding hydrogen atoms and missing heavy atoms, the entire system was solvated in a rectangular box by using TIP3P-Ewald water with a distance between the protein and the box boundary of 10.0 Å [41]. Counter ions Na^+^/Cl^−^ were added to the box to neutralize the system. Following minimization, both systems were heated from 0 K to 300 K for 50 ps. Next, a 10 ns simulation in the canonical ensemble was submitted to equilibration at 300 K. Then, the whole system was subjected to a 70 ns simulation in the NPT ensemble with a 2 fs time step until the RMSD of the systems converged. All molecular dynamics (MD) simulations were performed using the AMBER suite program [42] and Amber ff14SB force field [43]. Then, final snapshots of the MD simulations were used as the starting points for selecting the QM models [44,45]. In addition, the cluster model consists of the ATP molecule and the metal center Mg^2+^. Since previous work found that the coordination sphere of Mg^2+^ has a profound impact on the catalytic reaction [44,45], it is important to examine the coordination environment of Mg^2+^. In addition, All the boundary C atoms of the QM parts saturated with hydrogen atoms. And all the hydrogen atoms were added manually. The total numbers of atoms for Model-1, Model-2 and Model-3 are 169, 209 and 224, respectively. All the QM models have a net charge of +1e. To maintain the overall structure of the active site, the boundary carbon atoms and the captured hydrogen atoms were fixed at their corresponding positions in the crystal structure during geometry optimization.

### 4.2. Methods

The geometries of all the minima and transition states were determined by using the hybrid B3LYP [32,46,47,48] functional with the 6-31g (d, p) [49,50] basis set for all atoms. Vibrational frequencies were then obtained at the same level of theory to verify all the stationary points as local minima (zero imaginary frequencies) or first-saddle points (one imaginary frequency). IRC [51,52] calculations were also performed to identify the connection between transition states and minima. Based on the frequency calculations, the zero-point energy (ZPE) was also obtained. More accurate energies were further evaluated by the B3LYP functional with larger basis sets 6-311++g (d, p) for Mg^2+^ and cc-pVTZ for the rest. The van der Waals (VDW) effect was assessed by using Grimme’s D3 protocol [53]. The SMD model with a dielectric constant of 4 was used to mimic the protein environment during energy corrections [54,55]. All DFT calculations were implemented with the Gaussian09 software package [56].

To further study the exact role of the inhibitory glutamate in the AMPylation reaction of the HYPE protein, with the targeted BiP protein. We carried out MD simulations. Energy minimization and MD simulation were performed by using the sander module in the Amber14 program with the Amber ff14SB force field [57,58,59]. After 2000 steps of energy minimization for the entire system (1000 steepest descent steps and 1000 conjugate gradient steps), MD simulations were performed in the NPT ensemble at 1 bar pressure and a temperature of 300 K. The RMSD analysis and the RMSF analysis were carried out by the cpptraj module in the Amber14 program [57]. 

## Data Availability

The data presented in this study are available on request from the corresponding author.

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
