# Peer review of "Investigation of the Detailed AMPylated Reaction Mechanism for the Huntingtin Yeast-Interacting Protein E Enzyme HYPE"

_ijms, 2021, doi:10.3390/ijms22136999_

Round 1

Reviewer 1 Report

The manuscript by Liu and coworkers described a theoretical study about the enzymatic reaction catalyzed by the Huntingtin yeast-interacting protein E as AMP-transferring enzyme used for post-translational modification. My main appreciation of the manuscript is that maybe it would be more appropriate for other journal, rather than IJMS, and more devoted to theoretical studies.

I sincerely think that the text would need a strong improvement in organization and in the English language.

Point-to-point comments      

1.- The abstract needs to be reorganized to follow a clear though: general considerations, why to study this problem, and how the authors studied that.

2.- Figures need to be placed after their citation within the text

3.- Scheme 1: The authors would need to complete the legend, giving all the details regarding the meaning of each symbol. Moreover, what is the rationale to include such general scheme like that? Is this applied to the case of the AMP-transferring enzyme?

4.- The introduction, would also benefit from a strong organization. There is no clear trendline starting from the description of the scientific problem and the context. It can be exemplified by the lack of contextualization of the properties, characteristics, and importance of Fic proteins which are simply cited without carefully pointing them into a specific context. Abbreviations and acronyms need to be properly explained. The scientific names of any species should be written in italic (Line 46, Escherichia coli … and subsequent species). Probably it would be better to start talking about HYPE protein generically and then describe it in terms of structure and catalytic properties.

5.- Also in the introduction, the rationale for the use of the hybrid density functional theory method needs to be explained.

6.- Figure 1: the authors refer the figure as the representation of the different geometries of the active center of HYPE and its binding to ATP. However, the text started talking about the geometry of the binding of HYPE and BiP proteins. What is the rationale for the different models represented? Why the authors chose these different coordination geometries?

7.- Figure 2 needs to be increased in size to be visible. It is very difficult for the reader to differentiate all the legends

8.- The molecular dynamics part of the paper is correctly executed. In this part, the corresponding figures need to be improved, increasing the font size. Legends of the figures should include all the relevant information regarding the software used for the production of the Ligplot-like diagrams (Figure 3).

9.- The final section on computational details is also requiring a strong revision. Starting from the subsection 4.1 (Models), the authors should include more detailed information about the generation of the docking models of BiP and HYPE. It is not scientific to say that they choose the “reasonable docked structures”, and the information should be complemented with real data to be included in the supplementary section. It would be important to have the ZDock solutions ranked by their corresponding Gibbs functions. The protocol for MD simulations would need also further completion. The authors stated that they run MD simulations for 70 ns until RMSD converged, however the presented results in supplementary material did not seem to correspond to a converged system.

10.- In the Methods section (line 317 and following), the authors stated the use of “two geometries” for MD. In fact the authors are using a wild-type protein and a mutated protein Glu233Ala. The MD protocol in “Models” and “Methods” sections is somehow duplicated. Please organize the text in a more comprehensive manner.

11.- Figure 6: a more schematic representation of the protocol should be preferred. The use of 3D models does not help the reader in understanding what the authors really did.

Reviewer 2 Report

I have no comments

the manuscript can be accepted in current form

Author Response

Response: Thank you for reviewer's recognition and support for this work.

Reviewer 3 Report

This paper investigates the mechanism by which the HYPE protein ampilates the chaperone BiP. The authors performed DFT calculations and molecular dynamics simulations. The paper is well written and can be easily followed.

I recommend the following major improvements:

(1) One critical step for the mechanistic studies is the complex model of HYPE bound to BiP. Apparently, there is no experimental structural information on this complex. Recently, a related study was published in

https://link.springer.com/article/10.1007/s12192-021-01208-2

that also presented results from protein-protein docking. I request that Liu et al. discuss the general accuracy of protein-protein docking methods and also compare their results to this recent paper.

(2) The authors performed only a single molecular dynamics simulation of 70 ns in length. They find e.g. (line 201) that Arg374 has two interactions with O_gamma. Line 206: the average number of H-bonds is 9.3 for Ala234 and 5.2 with Glu234.

It is common practice to perform replicate MD simulations to address such questions and to prevent one-time observations. In the present case I would recommend to perform 3 MD simulations for each state that are started from the same starting structure with different random seeds. Then you can see if your findings are reproducible and you can give standard deviations for features such as H-bond numbers and really tell whether the differences are statistically significant.

(3) The authors used B3LYP calculations with the 6-31g (d,p) and 6-311g++g(d,p) / cc-pVTZ basis sets. They should comment on whether this functional and basis sets are suitable to describe suitable to describe an ampilyation-reaction. They could either show results for model reactions or cite some appropriate works from the literature for this.

(12) Figure S1: why are the electrostatic surface potentials so different in the 6-coordinated, 5-coordinated, and 4-coordinated models? Does this result from the identity of the amino acids included in the different models? I suspect that these strong differences could be an explanation for the different energetics that the authors reported in Fig. 1. So the different energetic profiles could simply result from the different coordination spheres included in the different models, but not necessarily reflect the coordination of Magnesium itself.

Further minor comments:

(5) p.2

line 18: correct AMPyltion

line 23: correct coordinated

line 29: with the -> while the

line 39: for typically -> and typically

second paragraph: use italic font for organisms

line 59: HYPE was -> HYPE is

(6) p.3

Line 78: found HYPE is the highly similar as VopS -> found that HYPE is highly similar to the VopS protein

Line 89: sequence mechanism -> sequential mechanism

Line 97: why is ref [32] cited here? That work seems not related to the present study.

Line 98: indicated -> indicate

Line 99: followed -> follows; was 4-coordinated -> is 4-coordinated

(7) p.4

Line 102: played -> plays

(8) p.5

Line 128: Why “in addition”?

(9) p.8

Line 217: RMSF fluctuations are shown in Figure S3, not S4.

(10) p.9

Line 242: Delete “We have”

Line 243: results revealed -> results suggest

Line 257: insert “was” in “glutamate was mutated”

(11) p.10

Line 279: One should treat results from protein-protein docking with some care. Therefore I suggest to modify this sentence into “First, 30 putative orientations of BiP docked to HYPE were obtained by Z-doc …”

Line 280: “According to geometry analysis and experimental information”: you should specify what were your criteria to select one of the docking models.

Line 296: “all the H-bonds”: which ones?

(12) p.10/11

Line 298: net charge of +1e.

Paragraph 4.1 Models: this paragraph mixes MD simulations and DFT calculations.

Actually, the two approaches seem unrelated. The DFT calculations use coordinates from the experimental structure of HYPE. How were Thr-518 and His363 positioned?

I recommend to split this paragraph into 2 paragraphs.

Figure 6: In the bottom panel, BiP looks smaller than in the top panel. The authors should explain in the figure legend why this is so and how BiP was cut.

(13) Figure S2 legend: something is wrong in “around 5 in complexes”.

(14) Figure S4 legend: add “involving ATP” after “Hydrogen Bonds”

Round 2

Reviewer 1 Report

I am happy with the answers to my questions and the modifications of the manuscript.

Reviewer 3 Report

The authors have fully addressed my minor points and have mostly addressed my 4 main points. However, I feel that their replies to my main points are not properly reflected by changes made to the main manuscript.

My previous point 1: I mentioned a recently published paper [57] that is actually available as preprint since 2018. The authors replied: "We have read it and cited it [57] in many places of our work, it enables our theoretical research to have more experimental verification."

Actually, I found only one citation to [57] in line 308 of the revised manuscript in the section "computational details". Given the relevance of that work to the present study, I request that the authors mention the approach and the results of [57] in the introduction of their manuscript where they reflect the current state of research.

My previous point 2 (single MD simulation): the authors have now added 2 further MD simulations of the same length. The only mentioning of this is in Fig. S6. Apparently, the final structures of the 3 simulations are quite similar. However, this is not quantified, e.g. in terms of RMSD. I request that the authors add a sentence in the methods section of the paper that they performed 3 replicate simulations and obtained similar results. Also, they should update Figs. S2 to S4 by including the results of the 2 additional simulations. My expectation is that the results will be quite similar to the ones for the included MD simulation, but this remains to be seen.

My previous point 3 (B3LYP): The authors have added refs 58-60 which is fine, but do not mention in the text why this is done. I have looked at refs 58 and 60. E.g. ref. 60 states "In previous studies, we have shown that the B3LYP functional was a good choice for phosphorus-containing systems". I request that the authors of this manuscript add a similar statement in the manuscript, e.g. in the methods section.

My previous point 4 (electrostatic potentials - different coordination spheres): The authors replied "We agreed with the reviewer’s opinion. The differences of the electrostatic surface potential could be an explanation for the different energetics in Figure 1. The different energetic profiles could simply result from the different coordination spheres included in different models; it is the mutual results of the interaction of surrounding amino acids cannot be simply said to be the problem of the coordination of Magnesium itself. We have corrected our understanding of this issue and the description in the manuscript."

I am glad that the authors agreed to my point. However, the only place in the revised manuscript, where the effect of coordination spheres is addressed, are lines 328-330 in the "methods section", where the authors state " Since previous work found that the coordination sphere of Mg2+ has a profound impact on the catalytic reaction [43,44], it is important to examine the coordination environment of Mg2+."

I request that the authors add several sentences in the discussion section of the manuscript in which they comment on the relationship between selecting different coordination spheres, including different amino acids, and the energetics of the reaction. In this context, also the connection to Fig. S1 should be mentioned.